# Smoking in social housing among adults in England, 2015–2020: a nationally representative survey

Sarah E Jackson ,[1] Hazel Cheeseman,[2] Deborah Arnott,[2] Robbie Titmarsh,[2] Jamie Brown[1]

¹Behavioural Science and Health, UCL, London, UK
²Action on Smoking and Health, London, UK

**Correspondence to**
Dr Sarah E Jackson;
s.e.jackson@ucl.ac.uk

## ABSTRACT

**Objectives** To analyse associations between living in social housing and smoking in England and to evaluate progress towards reducing disparities in smoking prevalence among residents of social housing compared with other housing types.

**Design** Cross-sectional analysis of nationally representative data collected between January 2015 and February 2020.

**Setting** England.

**Participants** 105 562 adults (≥16 years).

**Primary and secondary outcome measures** Linear and logistic regression were used to analyse associations between living in social housing (vs other housing types) and smoking status, cigarettes per day, time to first cigarette, exposure to others' smoking, motivation to stop smoking, quit attempts and use of cessation support. Analyses were adjusted for sex, age, social grade, region and year.

**Results** Adults living in social housing had two times the odds of being a smoker ($OR_{adj}$=2.17, 95% CI 2.08 to 2.27), and the decline in smoking prevalence between 2015 and 2020 was less pronounced in this high-risk group (−7%; $OR_{adj}$=0.98, 95% CI 0.96 to 1.01) than among adults living in other housing types (−24%; $OR_{adj}$=0.95, 95% CI 0.94 to 0.96; housing tenure–survey year interaction p=0.020). Smokers living in social housing were more addicted than those in other housing types (smoking within 30 min of waking: $OR_{adj}$=1.50, 95% CI 1.39 to 1.61), but were no less motivated to stop smoking ($OR_{adj}$=1.06, 95% CI 0.96 to 1.17) and had higher odds of having made a serious attempt to quit in the past year ($OR_{adj}$=1.16, 95% CI 1.07 to 1.25). Among smokers who had tried to quit, those living in social housing had higher odds of using evidence-based cessation support ($OR_{adj}$=1.22, 95% CI 1.07 to 1.39) but lower odds of remaining abstinent ($OR_{adj}$=0.63, 95% CI 0.52 to 0.76).

**Conclusions** There remain stark inequalities in smoking and quitting behaviour by housing tenure in England, with declines in prevalence stalling between 2015 and 2020 despite progress in the rest of the population. In the absence of targeted interventions to boost quitting among social housing residents, inequalities in health are likely to worsen.

## INTRODUCTION

Tobacco smoking is one of the leading drivers of health inequalities in England.[1] Higher

---

**STRENGTHS AND LIMITATIONS OF THIS STUDY**

⇒ A major strength of this study was the large sample, which was representative of adults living in England.

⇒ Another strength was the broad range of smoking outcomes assessed, offering a detailed view of smoking behaviour among people living in social housing compared with those living in other housing types.

⇒ The main limitation was that all outcomes were self-reported, introducing scope for bias.

---

smoking prevalence is associated with almost every indicator of socioeconomic disadvantage[2] and progress to reduce smoking prevalence has historically been slower among disadvantaged groups.[3 4] Understanding and alleviating this inequality is a priority for public health research and policy.

Housing tenure is an indicator of socioeconomic position that is particularly strongly linked with smoking.[5] In particular, social housing has been identified as a potential smoking 'hot spot'.[6] In England, social housing is let at lower rents on a secure, long-term basis to those who cannot afford to rent or buy a home on the open market, with priority given to those who have the greatest need. Accommodation is funded and regulated by the government and owned and managed by local authorities (local councils made up of publicly elected councillors) or housing associations (independent, not-for-profit organisations). A large survey in England in 2015–2017 revealed 34% of adults living in social housing were smokers, compared with 15% of people living in other housing types (eg, home owners or private renters).[6] Strikingly, smokers living in social housing were no less motivated to quit, but were only around half as likely to be successful when they tried.[6] This report prompted calls for targeted action to address this disparity.[7] The UK Government's 2017 tobacco control plan for England committed

to eliminating inequalities and reducing smoking prevalence in groups with the highest rates.[8] More recently, the Government committed to 'levelling up' disparities in health outcomes, incomes and educational opportunities.[9] What, if any, subsequent progress has been made in tackling smoking in social housing is unclear.

Using data from a nationally representative survey of more than 100 000 adults in England between 2015 and 2020, this study aimed to provide an update on smoking in social housing in England and evaluate progress towards reducing disparities in smoking prevalence among residents of social housing compared with other housing types.

## METHOD

### Design and population

Data were drawn from the Smoking Toolkit Study, a monthly cross-sectional survey representative of adults in England designed to provide insights into population-wide influences on smoking and cessation by monitoring trends on a range of variables relating to smoking.[10]

The Smoking Toolkit Study uses a hybrid of random probability and simple quota sampling to select a new sample of approximately 1700 adults aged≥16 years in England each month. To recruit each monthly sample, England is split into more than 170 000 output areas (consisting of approximately 300 households each). These output areas are stratified by Acorn characteristics (an established geodemographic analysis of the population; http://www.caci.co.uk/acorn/) and geographic region then randomly selected to be included in an interviewer's list. Interviewers travel to the selected areas and perform computer assisted interviews with one participant aged over 16 per household until quotas based on factors influencing the probability of being at home (working status, age and gender) are fulfilled. Participants complete a face-to-face computer-assisted survey with a trained interviewer. Comparisons with national data and cigarette sales indicate that key variables such as sociodemographic characteristics and smoking prevalence are nationally representative.[10 11]

Data on housing tenure, smoking and smoking cessation were collected between January 2015 and February 2020, so our analyses focus on participants recruited during this period. Data on housing tenure have not been collected since the COVID-19 pandemic required data collection to move from face-to-face to telephone interviews in March 2020, so these are the most up-to-date data available.

### Patient and public involvement (PPI)

The wider toolkit study has been discussed with a diverse PPI group, and the authors regularly attend and present at meetings at which patients and public are included. Interaction and discussion at these events help to shape the broad research priorities and questions. There is also a mechanism for generalised input from the wider public: each month interviewers seek feedback on the questions from all 1700 respondents, who are representative of the English population. This feedback is limited, and usually simply relates to understanding of questions and item options. No patients or members of the public were involved in setting the research questions or the outcome measures, nor were they involved in the design and implementation of this specific study. There are no plans to involve patients in dissemination.

### Measures

Housing tenure was categorised as 'social housing' (homes belonging to a housing association or rented from local authority; coded 1) versus 'other housing' (homes bought on a mortgage, owned outright, rented from private landlord or other; coded 0).

The smoking outcomes examined were:
1. Among all adults: cigarette smoking prevalence.
2. Among current smokers: mean number of cigarettes smoked per day (CPD) and percentage who smoke within 30 min of waking (as markers of cigarette dependence), high motivation to stop ('really want and plan to stop within 3 months'[12]) and regular exposure to smoking by others.
3. Among past-year smokers: percentage with a past-year quit attempt.
4. Among smokers with quit attempts in the past year: percentage not currently smoking, and who used cessation support (behavioural, nicotine replacement therapy over the counter, electronic cigarettes (e-cigarettes) or prescription medication).

Covariates were sex, age, occupational social grade (assessed using the National Readership Survey classification[13]), government office region and survey year.

### Statistical analysis

Data were analysed using SPSS V.27. Variables were weighted using rim (marginal) weighting to match an English population profile relevant to the time each monthly survey was conducted on the dimensions of age, social grade, region, housing tenure, ethnicity and working status within sex derived from English census data, Office for National Statistics mid-year estimates and other random probability surveys.[10] Missing data were removed on a per-analysis basis for each outcome.

We used linear regression (continuous outcomes) and logistic regression (binary outcomes) models to analyse associations between housing tenure (social housing vs other housing) and smoking outcomes, with and without adjustment for covariates. To test whether the effectiveness of use of evidence-based support for cessation differed by housing tenure, accounting for differences in dependence, we used logistic regression to test the interaction between housing tenure and use of evidence-based support, adjusting for covariates and measures of dependence (CPD and smoking within 30 min of waking).

Following peer review, we reran these analyses using log-binomial regression as an alternative to logistic

regression, to explore any differences in results. We also repeated our adjusted models with the inclusion of interactions between housing tenure and (1) age (16–34, 35–64 and ≥65 years) and (2) sex, to test for moderation of associations by these characteristics. Each interaction was tested in a separate model. Where interactions were statistically significant, we ran stratified analyses in which the association between housing tenure and the outcome variable was tested separately for each level of the moderating variable (ie, separately by age group or sex) to provide more information as to the nature of the differences between groups.

To examine differences in smoking prevalence trends by housing tenure over the study period, we graphically displayed annual data and reran the adjusted logistic regression model for smoking prevalence adding the interaction term between housing tenure and survey year (modelled as a continuous variable). We then ran stratified analyses in which the association between smoking prevalence and survey year was tested separately for each housing type (social vs other) to provide more information as to the nature of the difference between groups.

## RESULTS

A total of 105 562 adults aged≥16 years responded to the Smoking Toolkit Study survey between January 2015 and February 2020. Sample characteristics are shown in table 1. A total of 13 862 participants (13.1%) were social housing residents. Those living in social housing were more likely to be female, younger and from more disadvantaged social grades, and were more likely to live in London.

Associations between housing tenure and smoking outcomes are shown in table 2. Interactions between housing tenure and age group and sex are summarised in online supplemental tables 1 and 2, respectively, and stratified results are included in table 2 for outcomes where there was evidence of interaction with age or sex.

After adjustment for sex, age, social grade, region and survey year, adults living in social housing had more than double the odds of being a smoker compared with those living in other housing types. While this association was observed across all age groups and sexes, it was more pronounced among over 35s (vs 16–34 years) and women (vs men).

Current smokers living in social housing smoked on average one more cigarette per day and had 50% higher odds of smoking their first cigarette of the day within 30 min of waking, indicating significantly higher levels of addiction. These associations were strongest among younger adults (16–34 years), weaker among middle-aged adults (35–64 years) and were not statistically significant in the oldest group (≥65 years). Motivation to stop smoking did not differ significantly by housing tenure, nor did the odds of reporting regular exposure to smoking by others.

Smokers living in social housing had 16% higher odds of having made a serious attempt to quit in the past year

than those living in other housing types. Among smokers who had tried to quit in the past year, those living in social housing had 22% higher odds of using evidence-based cessation support (specifically, e-cigarettes or prescription medication). This difference was driven by smokers in the youngest age group (16–34 years), with no significant difference in use of support by housing tenure among middle-aged and older smokers. Despite greater use of support, smokers living in social housing had 37% lower odds of remaining abstinent after making a quit attempt. This does not mean evidence-based cessation support was less effective for smokers living in social housing: after adjustment for level of dependence, the association between use of evidence-based support and cessation did not differ significantly by housing tenure (interaction $OR_{adj}$ 0.93, 95% CI 0.64 to 1.34, p=0.684; online supplemental table 3).

There was little difference in the pattern of results when data were analysed using log-binomial regression (online supplemental table 4), although the difference in the rate of use of cessation support became non-significant ($RR_{adj}$ 1.09, 95% CI 0.99 to 1.21, p=0.086).

Figure 1 shows annual smoking prevalence estimates over the study period. There was a significant interaction between housing tenure and survey year on smoking prevalence ($OR_{adj}$ 1.03, 95% CI 1.01 to 1.06, p=0.020). Analyses stratified by housing tenure showed that there was a significant linear decline in smoking prevalence between 2015 and 2020 among adults living in other housing types ($OR_{adj}$ 0.95, 95% CI 0.94 to 0.96, p<0.001), with prevalence falling by 24% (from 16.0% in 2015 to 12.1% in 2020). However, the decline among adults living in social housing over the same period was not statistically significant ($OR_{adj}$ 0.98, 95% CI 0.96 to 1.01, p=0.120), falling by just 7% (from 35.3% in 2015 to 32.7% in 2020).

## DISCUSSION

This study extends the existing evidence base on smoking in social housing in England. Results showed adults who live in social housing remain more likely to smoke, with living in social housing particularly strongly linked to being a smoker in middle-aged and older adults and women. The general decline in smoking prevalence over recent years has stalled in this high-risk group compared with adults living in other housing types, indicating worsening inequalities in smoking on this measure. While smokers living in social housing are more addicted than those living in other housing (especially younger smokers), they are equally motivated to quit, more likely to make a quit attempt and more likely to use support. Yet they are less likely to be successful in stopping.

The results are consistent with those of a previous analysis that included data from 2015 to 2017,[6] suggesting there has been little change in smoking inequalities between adults who live in social vs other types of housing over recent years. A notable difference was that in this analysis, use of prescription medication as a cessation aid

**Table 1** Sample characteristics

| | Total (n=105 562) | | Social housing residents (n=13 862) | | Other housing residents (n=91 700) | |
|---|---|---|---|---|---|---|
| | n | % | n | % | n | % |
| Female | 53 830 | 51.0 | 8105 | 58.5 | 45 725 | 49.9 |
| Age (years) | | | | | | |
| 16–24 | 14 867 | 14.1 | 2101 | 15.2 | 12 766 | 13.9 |
| 25–34 | 17 744 | 16.8 | 2783 | 20.1 | 14 960 | 16.3 |
| 35–44 | 17 068 | 16.2 | 2300 | 16.6 | 14 768 | 16.1 |
| 45–54 | 18 190 | 17.2 | 2312 | 16.7 | 15 878 | 17.3 |
| 55–64 | 14 924 | 14.1 | 1739 | 12.5 | 13 185 | 14.4 |
| 65+ | 22 769 | 21.6 | 2626 | 18.9 | 20 142 | 22.0 |
| Social grade* | | | | | | |
| AB (most advantaged) | 28 649 | 27.1 | 719 | 5.2 | 27 930 | 30.5 |
| C1 | 29 420 | 27.9 | 2227 | 16.1 | 27 193 | 29.7 |
| C2 | 22 389 | 21.2 | 3351 | 24.2 | 19 038 | 20.8 |
| D | 15 742 | 14.9 | 3802 | 27.4 | 11 940 | 13.0 |
| E (most disadvantaged) | 9362 | 8.9 | 3764 | 27.2 | 5598 | 6.1 |
| Government office region | | | | | | |
| North East | 5181 | 4.9 | 887 | 6.4 | 4294 | 4.7 |
| North West | 13 915 | 13.2 | 1642 | 11.8 | 12 273 | 13.4 |
| Yorkshire and the Humber | 10 553 | 10.0 | 1193 | 8.6 | 9360 | 10.2 |
| East Midlands | 9164 | 8.7 | 1224 | 8.8 | 7940 | 8.7 |
| West Midlands | 10 850 | 10.3 | 1413 | 10.2 | 9437 | 10.3 |
| East of England | 11 851 | 11.2 | 1752 | 12.6 | 10 098 | 11.0 |
| London | 16 110 | 15.3 | 2782 | 20.1 | 13 328 | 14.5 |
| South East | 17 148 | 16.2 | 1733 | 12.5 | 15 415 | 16.8 |
| South West | 10 788 | 10.2 | 1235 | 8.9 | 9553 | 10.4 |
| Year of survey | | | | | | |
| 2015 | 19 988 | 18.9 | 2849 | 20.6 | 17 139 | 18.7 |
| 2016 | 20 433 | 19.4 | 2911 | 21.0 | 17 522 | 19.1 |
| 2017 | 20 395 | 19.3 | 2726 | 19.7 | 17 669 | 19.3 |
| 2018 | 20 703 | 19.6 | 2584 | 18.6 | 18 119 | 19.8 |
| 2019 | 20 641 | 19.6 | 2420 | 17.5 | 18 221 | 19.9 |
| 2020 | 3402 | 3.2 | 373 | 2.7 | 3029 | 3.3 |

*AB=managerial, administrative and professional; C1=supervisory, clerical and junior managerial, administrative and professional; C2=skilled manual workers; D semiskilled and unskilled manual workers; E=state pensioners, casual and lowest grade workers, unemployed with state benefits only.

was significantly higher among smokers living in social housing than other housing types when it had not been previously. This could be explained by a smaller reduction in use of prescription medication from the original to current analysis among smokers living in social housing (from 9.3% to 9.0%) than those living in other housing types (from 8.2% to 7.1%). It is encouraging that younger smokers in social housing were more likely to access evidence-based support, which can substantially increase their chances of quitting successfully, because their higher levels of dependence and various social and environmental barriers make it more difficult for them to successfully stop smoking. However, with four in ten quitters not using any form of evidence-based support, there remains room for improvement in helping smokers in social housing (and other housing tenures) to access effective support and translate more quit attempts into long-term cessation.

This analysis also provided some evidence of moderation of associations between housing tenure and smoking outcomes by age and sex. While living in social housing was associated with significantly higher odds of being a

**Table 2** Smoking and cessation behaviour among adults living in social housing compared with other housing, January 2015 to February 2020 (n=105 616), presented overall and stratified by age and sex where indicated by interactions

| | Social housing | Other housing | Unadjusted | | | Adjusted* | | |
|---|---|---|---|---|---|---|---|---|
| | | | OR/B† | 95% CI | P value | OR/B | 95% CI | P value |
| **All adults‡** | | | | | | | | |
| % cigarette smokers | 33.5 | 14.8 | 2.91 | 2.80 to 3.03 | <0.001 | 2.17 | 2.08 to 2.27 | <0.001 |
| Age 18–34 | 35.8 | 20.3 | 2.19 | 2.05 to 2.34 | <0.001 | 1.80 | 1.68 to 1.93 | <0.001 |
| Age 35–64 | 37.4 | 14.8 | 3.45 | 3.26 to 3.65 | <0.001 | 2.27 | 2.13 to 2.42 | <0.001 |
| Age≥65 | 19.7 | 7.1 | 3.20 | 2.87 to 3.57 | <0.001 | 2.58 | 2.29 to 2.90 | <0.001 |
| Male | 35.2 | 16.4 | 2.78 | 2.62 to 2.95 | <0.001 | 2.02 | 1.90 to 2.16 | <0.001 |
| Female | 32.3 | 13.1 | 3.16 | 2.99 to 3.33 | <0.001 | 2.31 | 2.18 to 2.45 | <0.001 |
| **Current cigarette smokers§** | | | | | | | | |
| Mean cigarettes per day | 12.2 | 10.5 | 1.72 | 1.45 to 1.99 | <0.001 | 0.97 | 0.69 to 1.25 | <0.001 |
| Age 18–34 | 11.0 | 8.8 | 2.20 | 1.82 to 2.58 | <0.001 | 1.50 | 1.11 to 1.90 | <0.001 |
| Age 35–64 | 13.0 | 11.6 | 1.38 | 0.98 to 1.78 | <0.001 | 0.68 | 0.26 to 1.11 | 0.002 |
| Age≥65 | 12.4 | 11.9 | 0.65 | −0.18 to 1.49 | <0.001 | 0.32 | −0.57 to 1.21 | 0.486 |
| % first smoke within 30 min of waking | 57.4 | 42.6 | 1.82 | 1.70 to 1.94 | <0.001 | 1.50 | 1.39 to 1.61 | <0.001 |
| Age 18–34 | 53.3 | 35.6 | 2.06 | 1.85 to 2.30 | <0.001 | 1.68 | 1.49 to 1.90 | <0.001 |
| Age 35–64 | 61.8 | 47.9 | 1.76 | 1.60 to 1.94 | <0.001 | 1.47 | 1.32 to 1.63 | <0.001 |
| Age≥65 | 50.8 | 45.6 | 1.23 | 1.01 to 1.51 | 0.042 | 1.12 | 0.90 to 1.39 | 0.309 |
| % high motivation to stop | 14.7 | 15.0 | 0.97 | 0.89 to 1.07 | 0.575 | 1.06 | 0.96 to 1.17 | 0.284 |
| % regular exposure to smoking by others | 68.4 | 68.6 | 0.99 | 0.92 to 1.06 | 0.778 | 1.01 | 0.94 to 1.10 | 0.749 |
| Age 18–34 | 73.5 | 76.3 | 0.86 | 0.76 to 0.97 | 0.016 | 0.94 | 0.82 to 1.08 | 0.380 |
| Age 35–64 | 67.0 | 65.8 | 1.06 | 0.96 to 1.17 | 0.279 | 1.05 | 0.94 to 1.17 | 0.415 |
| Age≥65 | 57.3 | 51.0 | 1.29 | 1.05 to 1.58 | 0.014 | 1.19 | 0.95 to 1.48 | 0.123 |
| **Past-year smokers¶** | | | | | | | | |
| % past-year quit attempt | 32.4 | 30.9 | 1.07 | 1.00 to 1.15 | 0.054 | 1.16 | 1.07 to 1.25 | <0.001 |
| **Past-year quit attempt**** | | | | | | | | |
| % not currently smoking | 11.6 | 18.9 | 0.56 | 0.47 to 0.67 | <0.001 | 0.63 | 0.52 to 0.76 | <0.001 |
| % used any cessation support | 59.0 | 54.4 | 1.20 | 1.07 to 1.35 | 0.002 | 1.22 | 1.07 to 1.39 | 0.003 |
| Age 18–34 | 56.0 | 47.4 | 1.41 | 1.17 to 1.68 | <0.001 | 1.43 | 1.17 to 1.74 | <0.001 |
| Age 35–64 | 61.9 | 61.2 | 1.03 | 0.87 to 1.22 | 0.731 | 1.05 | 0.87 to 1.27 | 0.591 |
| Age≥65 | 56.1 | 55.4 | 1.02 | 0.68 to 1.54 | 0.916 | 1.11 | 0.69 to 1.77 | 0.672 |
| % used behavioural support | 2.8 | 2.2 | 1.25 | 0.87 to 1.80 | 0.229 | 1.20 | 0.80 to 1.80 | 0.377 |
| % used NRT OTC | 13.4 | 13.0 | 1.04 | 0.88 to 1.23 | 0.671 | 0.88 | 0.73 to 1.07 | 0.189 |
| % used e-cigarettes | 33.9 | 32.1 | 1.08 | 0.96 to 1.23 | 0.196 | 1.19 | 1.04 to 1.36 | 0.012 |
| % used prescription medication | 9.0 | 7.1 | 1.28 | 1.04 to 1.58 | 0.020 | 1.33 | 1.05 to 1.68 | 0.017 |

Number of missing cases per variable: % cigarette smokers n=51 (0.0%); mean cigarettes per day n=325 (1.8%); % first smoke within 30 min of waking n=81 (0.4%); % high motivation to stop n=33 (0.2%); % regular exposure to smoking by others n=0 (0.0%); % past-year quit attempt n=556 (2.8%); % not currently smoking n=0 (0.0%); % used cessation support n=0 (0.0%).
Grey shading indicates results of subgroup analyses conducted when the interaction between housing tenure and age or sex (as relevant) was statistically significant.
*OR/B adjusted for sex, age, social grade, government office region and survey year.
†B can be interpreted as the mean (unadjusted/adjusted, as relevant) difference between the social housing and other housing groups.
‡All adults: social housing n=13 862; other housing n=91 700.
§Current cigarette smokers: social housing n=4637; other housing n=13 525.
¶Past-year smokers: social housing n=4923; other housing n=15 054.
**Past-year smokers who made a past-year quit attempt: social housing n=1551; other housing n=4530.
††Any cessation support includes behavioural support, nicotine replacement therapy (NRT) bought over the counter (OTC), e-cigarettes and prescription medication.

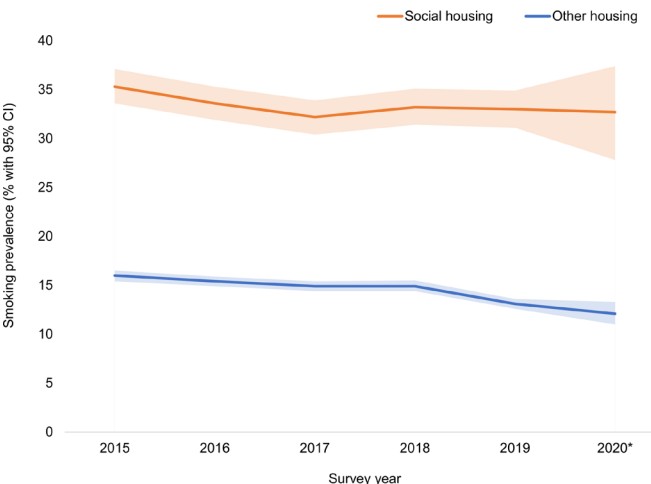

**Figure 1** Annual smoking prevalence among adults in England living in social housing compared with other housing tenures, January 2015 through February 2020. Shaded bands indicate 95% CIs. Bases (weighted n): social housing 2015 n=2849, 2016 n=2910, 2017 n=2717, 2018 n=2579, 2019 n=2420, 2020 n=373; other housing 2015 n=17 132, 2016 n=17 520, 2017 n=17 662, 2018 n=18 106, 2019 n=18 215, 2020 n=3029. *Data for 2020 are from January and February only.

smoker across men and women of all ages, this link was stronger among women compared with men. This may be an indication that women who live in social housing may be more likely than men to be experiencing other disadvantages (eg, being unemployed or a single parent), which compound their greater likelihood of smoking.[14] The disparity in smoking prevalence was also more pronounced among over-35s compared with those aged 16–34 years. In addition, the association between living in social housing and higher levels of addiction was strongest among the youngest age group (16–34 years), with no significant difference in level of addiction by housing tenure observed in the oldest age group (≥65 years). These findings suggest that living in social housing may be associated with greater risk of people who take up smoking at younger ages continuing to smoke throughout the life course. Younger adults who live in social housing are more likely to smoke than those who live in other housing types and, in particular, have higher levels of addiction, which make it harder for them to quit. This results in a greater disparity in smoking prevalence at older ages, as younger smokers outside social housing who have lower levels of addiction may quit with less difficulty before they reach middle age.

Without targeted action, smoking-related disparities are likely to have significant implications for the health of people and their families living in social housing. The adverse effects of smoking on health and life expectancy are well established, as is the transmission to the next generation,[15] but much of the harm caused by smoking can be reversed by quitting.[16 17] This offers huge policy potential to 'level up' and reduce the damage smoking causes. Various approaches have been suggested to

better support smokers in social housing, including ways in which social landlords can maximise their opportunity to improve tenants' well-being.[7] Most recently, the All Party Parliamentary Group on Smoking and Health recommended an at-scale intervention to provide free e-cigarettes and behavioural support to smokers in social housing[18] based on a successful pilot in Salford in the North of England.[19] We note that tobacco control measures often work synergistically and targeted policies are likely to be most effective in the context of a comprehensive, integrated approach.[18 20] Given the particularly high levels of addiction among younger smokers living in social housing and high prevalence of smoking at older ages, addressing uptake of smoking is an important target. Studies have shown that raising the age of sale can be effective in narrowing inequalities in initiation of smoking.[21 22]

A major strength of this study was the large, representative sample. There were also several limitations. First, all outcomes were self-reported, introducing scope for bias. Measurement of quit attempts and use of support relied on recall of the past year and quit success was not biochemically verified. While the latter would be a significant limitation in randomised trials (because smokers who receive active treatment may feel social pressure to claim abstinence), social pressure and the associated rate of misreporting is low in population surveys.[23] Moreover, we would not expect the extent of misreporting to differ by housing tenure meaning our results are unlikely to materially be affected. Second, while we adjusted for key sociodemographic variables, it is possible there was residual confounding by unmeasured variables, such as mental or physical health problems. Thirdly, the data were collected in England and the findings may not generalise to other countries with different approaches to social housing or tobacco control.

In conclusion, there remain stark inequalities in smoking and quitting behaviour by housing tenure in England, with declines in prevalence stalling between 2015 and 2020 despite progress in the rest of the population. In the absence of targeted interventions to boost quitting among social housing residents, inequalities in health are likely to worsen. In the context of the UK Government's commitment to levelling up, tackling smoking in social housing should be an urgent priority.

**Contributors** SEJ, HC, DA, RT and JB conceived and designed the study. SEJ analysed the data and wrote the first draft. All authors provided critical revisions. SEJ, as guarantor, accepts full responsibility for the work and/or the conduct of the study, had access to the data, and controlled the decision to publish.

**Funding** This work was supported by Cancer Research UK (C1417/A22962).

**Competing interests** JB has received unrestricted research funding from Pfizer, who manufacture smoking cessation medications. All authors declare no financial links with tobacco companies or e-cigarette manufacturers or their representatives.

**Patient and public involvement** Patients and/or the public were involved in the design, or conduct, or reporting, or dissemination plans of this research. Refer to the Method section for further details.

**Patient consent for publication** Not applicable.

**Ethics approval** This study involves human participants. Ethical approval for the STS was granted by the UCL Ethics Committee (ID 0498/001). The data are not collected by UCL and are anonymised when received by UCL. Participants gave informed consent to participate in the study before taking part.

**Provenance and peer review** Not commissioned; externally peer reviewed.

**Data availability statement** Data are available upon reasonable request. Data are available on request from the corresponding author.

**ORCID iD**
Sarah E Jackson http://orcid.org/0000-0001-5658-6168

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
