## [Reviewer comments · BMJ Open]

ARTICLE DETAILS

TITLE (PROVISIONAL)	Smoking in social housing among adults in England, 2015-2020: a nationally representative survey
AUTHORS	Jackson, Sarah; Cheeseman, Hazel; Arnott, Deborah; Titmarsh, Robbie; Brown, Jamie

VERSION 1 – REVIEW

REVIEWER	Ferrante, Gianluigi Azienda Ospedaliero Universitaria Città della Salute e della Scienza di Torino, Epidemiologia Screening
REVIEW RETURNED	03-Mar-2022

GENERAL COMMENTS	This paper, based on a representative cross-sectional study of the adult population in England, investigates the association between housing tenure (living in social housing vs. living in other housing type) and a number of smoking outcomes. The findings show that there are still large inequalities in smoking and quitting behaviour related to living in social housing, identifying a subgroup of the population that should be prioritised for health promotion interventions of smoking cessation. The manuscript is well written and represents an important piece of knowledge for guiding a certain type of public health interventions. The work is well organised, has clear and concise language. I have no particular comments to make, other than to highlight a methodological aspect of the analysis that could at least be explored in advance before publishing the work. The authors used logistic regression models to analyse associations between housing tenure and smoking outcomes, adjusting for covariates. In some cases, the odds ratio can importantly overestimate the prevalence ratio, the measure of choice in cross-sectional studies. I do not know if this is the case for your study, but before giving the ok for publication I would like to ask you to repeat the analyses for binary outcomes using models with robust variance estimators in Poisson regressions. If the results of the latter models do not differ from those estimated through logistic regressions, I would proceed with publication. If not, I would ask you to use the models suggested for the analyses in Table 2. Please find below a bibliographic reference to support what I am asking you to try. Bibliographic reference
--

	Barros AJ, Hirakata VN. Alternatives for logistic regression in cross-sectional studies: an empirical comparison of models that directly estimate the prevalence ratio. BMC Med Res Methodol. 2003 Oct 20;3:21. doi: 10.1186/1471-2288-3-21. PMID: 14567763; PMCID: PMC521200.
--	---

REVIEWER	Li, Lihua Icahn School of Medicine at Mount Sinai, Population health sciences and policy
REVIEW RETURNED	14-Mar-2022

GENERAL COMMENTS	Thank you for the opportunity to review the manuscript titled “Smoking in social housing among adults in England, 2015-2020: a nationally representative survey”. This manuscript examined the associations between living in social housing and smoking in England and evaluated progress toward reducing disparities in smoking prevalence among residents of social housing compared with other housing types. The topic is relevant though not much innovative, but the paper can be improved in many aspects, particularly on analysis. Below, I discuss a few major concerns, followed by a few minor points about the paper. Major  1. There are very few characteristics variables collected, which may result in residual confounding in the analysis. Other than age, social grade and region, other factors including participants’ race, employment status, history and intensity of smoking, health condition, comorbidity and access to health care are well studied to be associated with their intent and attempt to quit smoking, as well quit behavior. For example, people with more comorbidities are usually more likely to quit smoking. However, these important factors were not considered. I would suggest including more covariates in the adjusted regression models. 2. The authors claimed that a Nationally-representative, cross-sectional survey between January 2015 and February 2020 was used. However, it was not clear what the survey is about, how the survey is designed and how the analysis accounted for the survey design. For instance, did the analysis consider the survey design elements, such as weight? Was strata and cluster involved? Was the regression survey weighted? I would suggest that the author add the description of the survey and survey methods, and consider survey weighted regression analysis. 3. The smoking patterns and quitting behaviors differs by age, such as adults and old adults and youth. In this paper, the survey participants were pooled for analysis, with age ranged from 16 to 65 and above. I would suggest adding subgroup analysis to examine this association in a granular way by stratifying the participants by age. 4. There is large room for the description of results section to be improved. For example, there was no description of table 1 except saying “Sample characteristics are show in table 1”. Also it was not reported that how many participants were included each year. How many subjects with missing value and how the missing value was handled were not clear.
--

	5. Discussion could also be expanded by not just restricting the target population to those living in the social housing. As housing type may just serve as another measure or indicator of socio-economics status. The target intervention should be considered in the setting of an integrated mechanism. Minor:  1. Design, setting and participants sections in the abstract were poorly described. For example, the “Nationally-representative, cross-sectional survey between January 2015 and February 2020” is the description of survey design, but the study design. 2. In the results section, “the association between use of evidence-based support and cessation did not differ significantly by housing tenure (interaction ORadj 0.93, 95% CI 0.64-1.34, p=0.684)” was not reported in the tables. The output with interactions included should be included as supplementary materials. 3. Smoking and quitting behavior may also differ by gender. I would suggest testing the interaction with gender too.
--	---

REVIEWER	Karadoğan, Dilek Recep Tayyip Erdoğan University
REVIEW RETURNED	15-Mar-2022

GENERAL COMMENTS	The study with a large sample size evaluates the smoking characteristics of two group, 1. social housing and 2. other housing. In general the differences of the adults in that places, and also the differences between this places are not clear. And therefore without describing effectors clearly, this research question and the provided findings are not relevant, in my opinion.
---

VERSION 1 – AUTHOR RESPONSE

Reviewer: 1

Dr. Gianluigi Ferrante, Azienda Ospedaliero Universitaria Città della Salute e della Scienza di Torino

This paper, based on a representative cross-sectional study of the adult population in England, investigates the association between housing tenure (living in social housing vs. living in other housing type) and a number of smoking outcomes.

The findings show that there are still large inequalities in smoking and quitting behaviour related to living in social housing, identifying a subgroup of the population that should be prioritised for health promotion interventions of smoking cessation.

The manuscript is well written and represents an important piece of knowledge for guiding a certain type of public health interventions.

The work is well organised, has clear and concise language. I have no particular comments to make, other than to highlight a methodological aspect of the analysis that could at least be explored in advance before publishing the work.

The authors used logistic regression models to analyse associations between housing tenure and smoking outcomes, adjusting for covariates. In some cases, the odds ratio can importantly overestimate the prevalence ratio, the measure of choice in cross-sectional studies.

I do not know if this is the case for your study, but before giving the ok for publication I would like to ask you to repeat the analyses for binary outcomes using models with robust variance estimators in Poisson regressions. If the results of the latter models do not differ from those estimated through logistic regressions, I would proceed with publication. If not, I would ask you to use the models suggested for the analyses in Table 2. Please find below a bibliographic reference to support what I am asking you to try.

Response: We have added a sensitivity analysis using log-binomial regression rather than logistic regression (as supported by the reference provided by the reviewer). There was little difference in the pattern of results, so we have retained the logistic regression results for our primary analysis and present the log-binomial regression results in Supplementary Table 4.

Bibliographic reference

Barros AJ, Hirakata VN. Alternatives for logistic regression in cross-sectional studies: an empirical comparison of models that directly estimate the prevalence ratio. *BMC Med Res Methodol.* 2003 Oct 20;3:21. doi: 10.1186/1471-2288-3-21. PMID: 14567763; PMCID: PMC521200.

Reviewer: 2

Dr. Lihua Li, Icahn School of Medicine at Mount Sinai

Thank you for the opportunity to review the manuscript titled “Smoking in social housing among adults in England, 2015-2020: a nationally representative survey”. This manuscript examined the associations between living in social housing and smoking in England and evaluated progress toward reducing disparities in smoking prevalence among residents of social housing compared with other housing types. The topic is relevant though not much innovative, but the paper can be improved in many aspects, particularly on analysis. Below, I discuss a few major concerns, followed by a few minor points about the paper.

Major

1. There are very few characteristics variables collected, which may result in residual confounding in the analysis. Other than age, social grade and region, other factors including participants’ race, employment status, history and intensity of smoking, health condition, comorbidity and access to health care are well studied to be associated with their intent and attempt to quit smoking, as well quit behavior. For example, people with more comorbidities are usually more likely to quit smoking. However, these important factors were not considered. I would suggest including more covariates in the adjusted regression models.

Response: We agree that variables such as comorbid health conditions are relevant. Unfortunately, the Smoking Toolkit Study did not collect this information as part of the survey across all the relevant waves. We now mention the possibility of residual confounding in the discussion of the study’s limitations:

“Secondly, while we adjusted for key sociodemographic variables, it is possible there was residual confounding by unmeasured variables, such as mental or physical health problems.”

While we did consider including other measures of socioeconomic position, such as employment status, we decided against it due to the high degree of multicollinearity with social grade and housing tenure this would introduce to the models. Previous research has identified social grade to be the strongest predictor of smoking status, after housing tenure, in this population (<https://academic.oup.com/ntr/article/23/1/107/5728574>).

2. The authors claimed that a Nationally-representative, cross-sectional survey between January 2015 and February 2020 was used. However, it was not clear what the survey is about, how the survey is designed and how the analysis accounted for the survey design. For instance, did the analysis consider the survey design elements, such as weight? Was strata and cluster involved? Was

the regression survey weighted? I would suggest that the author add the description of the survey and survey methods, and consider survey weighted regression analysis.

Response: We now provide a more detailed description of the survey design and methodology:

“Data were drawn from the Smoking Toolkit Study, a monthly cross-sectional survey representative of adults in England designed to provide insights into population-wide influences on smoking and cessation by monitoring trends on a range of variables relating to smoking (10).

“The Smoking Toolkit Study uses a hybrid of random probability and simple quota sampling to select a new sample of approximately 1,700 adults aged ≥ 16 years in England each month. To recruit each monthly sample, England is split into more than 170,000 output areas (consisting of approximately 300 households each). These output areas are stratified by ACORN characteristics (an established geo-demographic analysis of the population; <http://www.caci.co.uk/acorn/> and geographic region then randomly selected to be included in an interviewer's list. Interviewers travel to the selected areas and perform computer assisted interviews with one participant aged over 16 per household until quotas based upon factors influencing the probability of being at home (working status, age, and gender) are fulfilled. Participants complete a face-to-face computer-assisted survey with a trained interviewer. Comparisons with national data and cigarette sales indicate that key variables such as sociodemographic characteristics and smoking prevalence are nationally representative (10,11).”
All analyses are weighted to match the population in England, as described in the statistical analysis section:

“Variables were weighted using rim (marginal) weighting to match an English population profile relevant to the time each monthly survey was conducted on the dimensions of age, social grade, region, housing tenure, ethnicity and working status within sex derived from English census data, ONS mid-year estimates and other random probability surveys (10).”

3. The smoking patterns and quitting behaviors differs by age, such as adults and old adults and youth. In this paper, the survey participants were pooled for analysis, with age ranged from 16 to 65 and above. I would suggest adding subgroup analysis to examine this association in a granular way by stratifying the participants by age.

Response: We have added tests of interactions between housing tenure and age. These suggest the link between living in social housing and being a smoker is stronger among middle-aged and older adults ($\geq 35y$) compared with younger adults (16-34y; although the association is significant in all age groups). On the other hand, the association between housing tenure and level of addiction is more pronounced in the younger group, with no significant difference in addiction among those aged $\geq 65y$. We have also added a section in the discussion that offers some reflection on these results.

4. There is large room for the description of results section to be improved. For example, there was no description of table 1 except saying “Sample characteristics are show in table 1”.

Response: We now provide a brief summary of the content of Table 1 in the text:

“A total of 13,862 participants (13.1%) were social housing residents. Those living in social housing were more likely to be female, younger, and from more disadvantaged social grades, and were more likely to live in London.”

Also it was not reported that how many participants were included each year.

Response: We have added details on the number of participants surveyed each year to Table 1.

How many subjects with missing value and how the missing value was handled were not clear.

Response: As there was a very small amount of missing data, cases were removed on a per-analysis basis for each outcome (reported in the method under ‘Statistical analysis’). The number of cases with missing data on each variable of interest is reported in the footnote under Table 2:

“Number of missing cases per variable: % cigarette smokers $n=51$ (0.0%); mean cigarettes per day $n=325$ (1.8%); % first smoke within 30 min of waking $n=81$ (0.4%); % high motivation to stop $n=33$ (0.2%); % regular exposure to smoking by others $n=0$ (0.0%); % past year quit attempt $n=556$ (2.8%); % not currently smoking $n=0$ (0.0%); % used cessation support $n=0$ (0.0%)”.

5. Discussion could also be expanded by not just restricting the target population to those living in the social housing. As housing type may just serve as another measure or indicator of socio-economics status. The target intervention should be considered in the setting of an integrated mechanism.

Response: We have added the following to the discussion to make this point:

“We note that tobacco control measures often work synergistically and targeted policies are likely to be most effective in the context of a comprehensive, integrated approach (17,19).”

Minor:

1. Design, setting and participants sections in the abstract were poorly described. For example, the “Nationally-representative, cross-sectional survey between January 2015 and February 2020” is the description of survey design, but the study design.

Response: We have reworded to read: “Cross-sectional analysis of nationally-representative data collected between January 2015 and February 2020.”

2. In the results section, “the association between use of evidence-based support and cessation did not differ significantly by housing tenure (interaction ORadj 0.93, 95% CI 0.64-1.34, p=0.684)” was not reported in the tables. The output with interactions included should be included as supplementary materials.

Response: We now report the main effects and interaction for this analysis in Supplementary Table 3.

3. Smoking and quitting behavior may also differ by gender. I would suggest testing the interaction with gender too.

Response: We have added tests of interactions between housing tenure and gender. These suggest the link between living in social housing and being a smoker is stronger among women than men (although the association is significant in both groups).

Reviewer: 3

Dr. Dilek Karadoğan, Recep Tayyip Erdoğan University

The study with a large sample size evaluates the smoking characteristics of two group, 1. social housing and 2. other housing.

In general the differences of the adults in that places, and also the differences between this places are not clear. And therefore without describing effectors clearly, this research question and the provided findings are not relevant, in my opinion.

Response: We have added an explanation of social housing to the introduction:

“In England, social housing is let at lower rents on a secure, long-term basis to those who cannot afford to rent or buy a home on the open market, with priority given to those who have the greatest need. Accommodation is funded and regulated by the government and owned and managed by local authorities (local councils made up of publicly elected councillors) or housing associations (independent, not-for-profit organisations).”

Table 1 provides a summary of the sociodemographic characteristics of people living in social housing compared with other housing types in England.

VERSION 2 – REVIEW

REVIEWER	Li, Lihua Icahn School of Medicine at Mount Sinai, Population health sciences and policy
REVIEW RETURNED	29-Jun-2022
GENERAL COMMENTS	The authors have addressed my concerns. The manuscript has improved significantly after its revision. One suggestion is that

	reporting effect by each subgroup would be more meaningful than reporting the interaction effect.
--	---